# Immunomodulatory Potency of *Eclipta alba* (Bhringaraj) Leaf Extract in *Heteropneustes fossilis* against Oomycete Pathogen, *Aphanomyces invadans*

**DOI:** 10.3390/jof9020142

**Published:** 2023-01-21

**Authors:** Vikash Kumar, Basanta Kumar Das, Himanshu Sekhar Swain, Hemanta Chowdhury, Suvra Roy, Asit Kumar Bera, Ramesh Chandra Malick, Bijay Kumar Behera

**Affiliations:** 1Aquatic Environmental Biotechnology and Nanotechnology (AEBN) Division, ICAR-Central Inland Fisheries Research Institute (CIFRI), Barrackpore 700120, India; 2ICAR-Central Inland Fisheries Research Institute (CIFRI), Barrackpore 700120, India; 3Fisheries Enhancement & Management (FEM) Division, ICAR-Central Inland Fisheries Research Institute (CIFRI), Barrackpore 700120, India; 4Reservoir & Wetland Fisheries (RWF) Division, ICAR-Central Inland Fisheries Research Institute (CIFRI), Barrackpore 700120, India; 5Fisheries Resource Assessment & Informatics (FRAI) Division, ICAR-Central Inland Fisheries Research Institute (CIFRI), Barrackpore 700120, India

**Keywords:** *Aphanomyces invadans*, *Eclipta alba* leaf extract, anti-stress, antioxidant, immune response

## Abstract

*Aphanomyces invadans* is an aquatic oomycete pathogen and the causative agent of epizootic ulcerative syndrome (EUS) in fresh and brackish water fish, which is responsible for severe mortalities and economic losses in aquaculture. Therefore, there is an urgent need to develop anti-infective strategies to control EUS. An Oomycetes, a fungus-like eukaryotic microorganism, and a susceptible species, i.e., *Heteropneustes fossilis*, are used to establish whether an *Eclipta alba* leaf extract is effective against the EUS-causing *A. invadans*. We found that treatment with methanolic leaf extract, at concentrations between 50–100 ppm (T4–T6), protects the *H. fossilis* fingerlings against *A. invadans* infection. These optimum concentrations induced anti-stress and antioxidative response in fish, marked by a significant decrease in cortisol and elevated levels of superoxide dismutase (SOD) and catalase (CAT) levels in treated animals, as compared with the controls. We further demonstrated that the *A. invadans*-protective effect of methanolic leaf extract was caused by its immunomodulatory effect and is linked to the enhanced survival of fingerlings. The analysis of non-specific and specific immune factors confirms that methanolic leaf extract-induced HSP70, HSP90 and IgM levels mediate the survival of *H. fossilis* fingerlings against *A. invadans* infection. Taken together, our study provides evidence that the generation of anti-stress and antioxidative responses, as well as humoral immunity, may play a role in protecting *H. fossilis* fingerlings against *A. invadans* infection. It is probable that *E. alba* methanolic leaf extract treatment might become part of a holistic strategy to control EUS in fish species.

## 1. Introduction

*Aphanomyces invadans*, an aquatic oomycete most frequently recognized as a causative agent of the epizootic ulcerative syndrome (EUS) is a seasonal epidemic pathogen of great importance in wild and farmed fish in both freshwater and estuarine environments [1,2,3,4]. EUS is an OIE-listed disease, causing complex infectious etiology, which leads to necrosis ulcerative lesions and granulomatous response in fishes. The disease was first identified from Japan in 1971, and later it was reported in south-east Queensland (Australia) in 1972. Subsequently, the outbreaks spread widely across North America and the Asia-Pacific region [5]. It is one of the most destructive diseases among freshwater and brackish water fish in the Asia-Pacific region, and about 125 fish species have been confirmed to be affected by EUS, causing a huge economic impact. For instance, during the late 1980s and early 1990s, it caused an estimated loss of USD 110 million in the aquaculture farms of the Asia-Pacific region [6,7]. 

The traditional methods applied so far in the mitigation or cure of microbial infection, such as disinfectants and antibiotics, had very limited success. Additionally, their application in the food production sector is under severe public and scientific scrutiny due to the development of multiple antibiotic-resistant microbial phenotypes. This created a huge concern among food production and drug development regulatory bodies, and more emphasis has been given in the development of plant-based and/or natural compounds to enhance the immune reactivity and disease resistance in fish against pathogenic microbes [8,9]. It is important to mention that several environmental and biological factors are responsible for the growth and establishment of *A. invadans*, which further attracts secondary pathogens to enter the lesions, thus increasing the severity of the infection [10,11,12,13]. Hence, a holistic approach focuses on the development of plant-derived compounds or natural products that enhance the host immune system and generate anti-infective strategies that will have the highest chance of decreasing the risk of pathogenic *A. invadans* infection in fish.

The use of plant-based compounds or natural products has attracted keen interest in the past decades due to their antimicrobial, immunostimulant and antioxidant properties. For instance, Harikrishnan et al. [14,15] demonstrated that dietary supplementation of plant-based compounds significantly improves immune response and disease resistance in fish infected with the fungal pathogen, *A. invadans*. Afzali and Wong [16] reported that dietary supplementation of the *Sonneratia alba* extract enhances immunity and provides protection and disease resistance in goldfish against *A. invadans*. Kumar et al. [17,18] further highlighted that plant-based compounds (*Mikania cordata* leaf powder) can be used as a potential immunostimulatory agent to enhance the non-specific immune response and disease resistance of fish against the *A. invadans* infection. Although these studies have shown promising results in enhancing immune response and developing resistance in fish, there was no further work to utilize this management approach to control *A. invadans* infection. The *Eclipta alba* belongs to the family Asteraceae and is commonly known as false daisy in English and bhringraj in India; it has several ethnomedicinal properties. The plant has diverse medicinal values mainly used for the treatment of gastrointestinal disorders, respiratory tract disorders, fever, hair loss and graying of hair, liver disorders, skin disorders, spleen enlargement and cuts and wounds in higher vertebrates, including humans [19]. This motivated us to investigate the potential role of *E. alba* leaf extract against a commercially important Oomycetes disease, epizootic ulcerative syndrome (EUS). The study describes that *E. alba* leaf methanolic extract induces anti-stress, antioxidant and non-specific and specific immunity that leads to increased survival of *H. fossilis* fingerlings against EUS-causing *A. invadans*. 

In this study, using an Oomycetes, a fungus-like eukaryotic pathogen, and a susceptible fish species, i.e., *Heteropneustes fossilis*, we aimed to investigate whether the methanolic extract of *E. alba* leaves potentiates the generation of in vivo anti-stress, antioxidant and immunity. Hereto, the *H. fossilis* fingerlings were exposed to methanolic leaf extract of *E. alba* and Oomycetes pathogen *A. invadans*. Underlying protective mechanisms were examined by studying the antioxidant enzyme activity, serum biochemical indices and immune-stress responses. To the best of our knowledge, this is the first description of *E. alba* leaves’ methanolic extract-induced immunomodulation, resulting in the induction of resistance in *H. fossilis* fingerlings against pathogenic *A. invadans*.

## 2. Materials and Methods

### 2.1. Oomycetes Isolate Used in the Study

*Aphanomyces invadans* zoospores for fish challenge experiments were collected from the aquatic environmental biotechnology and nanotechnology (AEBN) division, ICAR-central inland fisheries research institute (CIFRI), Barrackpore, India. Following OIE (2003) guidelines with minor modifications, the *A. invadans* isolate was grown in potato dextrose agar (PDA) plates (HiMedia, Mumbai, India). For spore production, *A. invadans* mycelium was allowed to grow in potato dextrose broth (PDB) (HiMedia, Mumbai, India) at 20 °C for 5 days (Figure 1). The culture was washed 3 times and resuspended in autoclaved pond water (APW). The suspension was then placed in 2 mL Eppendorf tubes containing APW with 0.05% CaCO3 and left for 24 h. The number of zoospores in the suspension was calculated by using a Neubauer counting chamber and was adjusted to the required concentration. 

### 2.2. PCR Confirmation of A. invadans Isolate

The Oomycetes genomic DNA was isolated following the protocol of the Sarkosyl method (Sambrook and Russel, 2001). The quality of extracted DNA was checked in 1% agarose gel and quantified using Nanodrop (Eppendorf, Hamburg, Germany). The ITS1 primer including forward and reverse primers were used for amplification- and confirmation-specific site of *A. invadans* isolate located in the ITS1 region using thermal cycler Gene Amp PCR system 9700 (Applied Biosystems, Foster City, CA, USA) (Table 1). The PCR mixture consisted of 25 pM of each primer, deoxynucleoside triphosphate at the concentration of 2.5 mM, 0.5 units of Taq DNA polymerase and 20 ng of *A. invadans* genomic DNA for a total of 50 µL reaction volume to the following cyclic condition: 95 °C for 2 min, 35 cycles of 30 s at 95 °C each, 56 °C for 45 s, 72 °C for 2.5 min and 72 °C at 5 min for the final extension. The agarose gel electrophoresis analyzes the PCR product and the target product is 234 bp (Figure 2) (Vandersea et al. 2006). The PCR products are sequenced in forward and reverse direction and contig was prepared by aligning forward and reverse sequences using DNA baser 7.0.0. Afterward, the sequence was compared with available sequences in species of Oomycetes and fungal family in the NCBI GenBank with BLAST (http://blast.ncbi.nlm.nih.gov) accessed on 14 June 2022.

### 2.3. Isolation of Methanolic Leaf Extract from Eclipta alba (Bhringaraj)

*E. alba* (Bhringaraj) is a herbaceous plant that grows wildly in moist places as a weed and has numerous medicinal properties (Figure 3). The leaves extracts were reported to contain bioactive compounds, viz., Stigmasterol, β-terthienylmethanol, wedelolactone, demethyl wedelolactone and dimethyl wedelolactone-7-glucoside, having several beneficial therapeutic properties (Figure 4) (Wagner et al., 1986). For the study, the leaves (2 kg) were collected from the experimental farm of ICAR-Central Inland Fisheries Research Institute, Barrackpore, India. Initially, the plant samples were washed properly with fresh water to remove the dust and soil particles and then dried under shade. The dried leaf samples were then powdered in a mixer grinder. The leaf powder was initially extracted in a Soxhlet apparatus for 8 h in petroleum-ether solvent (leaf and solvent ratio, 1:2). The suspension in the flask is kept for 3 days, and after every 8 h the solution was mixed thoroughly. Afterward, the solution was filtered and pet-ether extract was discarded. The content was dried completely in the shade. The residue was added with 100 % methanol and kept for 3 days. Similarly, the suspension in the flask was mixed thoroughly every 8 h. After 3 days, the solution was filtered and kept in a glass bottle at 4 °C. The methanolic extract was evaporated in a rotary evaporator to until dry. This crude methanolic extract (~6 mL) was stored in a glass bottle at 4 °C until further used for analysis.

### 2.4. Eclipta alba (Bhringaraj) Leaf Methanolic Extract Lethality Test and Challenge Assay

#### 2.4.1. Ethical Approval

Organization for Economic Cooperation and Development (OECD) guidelines were followed for the handling and care of experimental animals. The animal utilization protocol was approved by Institutional Animal Ethics Committee, ICAR-Central Inland Fisheries Research Institute, Kolkata, India, (IAEC/2021/04) for the experimental setup.

#### 2.4.2. Acclimatization of Experimental Fish

Healthy *Heteropneustes fossilis* (Stinging catfish) of size (length = 100.4 ± 5.3 mm, weight = 15.83 ± 2.8 g) were collected from ICAR-CIFRI, Barrackpore, India culture facility. There were no external clinical symptoms, such as hemorrhage, ulcer, discoloration, descaling and redness, in the body of the fish. Additionally, before the challenge assay, the fish were randomly selected and screened for the presence of possible infectious microbes following the standard protocol (Nickum et al., 2004; Johansen et al., 2006). The fish were acclimatized in 200 L FRP tanks for 2 weeks and commercial floating feed (CP: 30%, CL: 5%) was fed to fish at 5–3% of the biomass in two equal installments (10:00 h, 15.00 h).

#### 2.4.3. Lethality Test of *E. alba* Extract

In the first experiment, the cytotoxic effect of *E. alba* (Bhringaraj) methanolic leaf extract was determined on the *H. fossilis*, as described previously [20] with slight modifications. Briefly, 20 nos. of *H. fossilis* fingerlings were randomly in 40 liters of glass aquaria containing 20 liters of water. The fingerlings were exposed to increasing concentrations of methanolic leaf extract, viz., 5 (T1), 10 (T2), 25 (T3), 50 (T4), 75 (T5), 100 (T6) and 125 ppm (T7) in 1% of dimethyl sulfoxide (DMSO) for 168 h. The *H. fossilis* fingerlings group that did not receive the Bhringaraj leaf extract but supplemented with only 1% of DMSO served as a control. The toxicity of the compound was determined 168 h post-exposure by scoring the number of survivors, as previously described [21]. Three replicates were maintained for each treatment and control group. 

#### 2.4.4. LC_50_ Analysis of *A. invadans*

In the second experiment, it was determined whether *A. invadans* isolate could cause mortality in *H. fossilis* fingerlings, and to this end, an LC_50_ study was carried out. Following OIE (2019) bioassay protocol described for fish susceptible to infection with *A. invadans*-susceptible species (e.g., *H. fossilis*), serial dilutions of five test concentrations of *A. invadans* @ 0.5, 1, 2, 4, and 8 mL suspension of 100+ motile zoospores/liter of water were prepared to evaluate the Oomycetes lethal concentration required for mortality of 50% (LC50) of the fish population. The number of zoospores in the suspension was calculated by using a Neubauer counting chamber and was adjusted to the required concentration. The *H. fossilis* fingerlings (20 nos.) were randomly distributed in six different tanks; one tank for control and five tanks for challenge study. The fingerlings were exposed to increasing concentrations of *A. invadans*, viz., 0.5, 1, 2, 4, and 8 mL suspension of 100+ motile zoospores/liter of water at 20 °C. The control group of fish was added to a sterilized normal saline solution. The survival of *H. fossilis* fingerlings was scored 168 h after the addition of the pathogen, and each treatment was performed in triplicate.

#### 2.4.5. Dose Response Study

In the next experiment, independent challenge assays were performed to determine the dose–response relationship (protective effect) of *E. alba* methanolic leaf extract. Firstly, 40 µL of *A. invadans* spore suspension in potato dextrose broth (PDB) (HiMedia, Mumbai, India) was inoculated onto 20 mL of sterile PDB in 50 mL Erlenmeyer flask. The culture was incubated for 120 h at 20 °C. The zoospores number in the suspension was calculated by using a Neubauer counting chamber and was adjusted to the required concentration. Afterward, a group of 20 nos. of *H. fossilis* fingerlings was randomly distributed in 40 liters of glass aquaria containing 20 liters of water. The aquaria were supplemented with increasing concentrations of methanolic leaf extract (non-lethal dose), viz., 5 (T1), 10 (T2), 25 (T3), 50 (T4), 75 (T5), and 100 ppm (T6) in 1% of DMSO for 168 h, as described above in the toxicity assay. Simultaneously, the aquaria were added separately with *A. invadans* @ 4 mL suspension of 100+ motile zoospores/liter of water at 20 °C (based on LC_50_ analysis of *A. invadans*). The survival of *H. fossilis* fingerlings was scored 168 h after the addition of the pathogen. The non-exposed fingerlings supplemented with only 1% of DMSO (Bhringaraj leaf extract) that were either challenged or not with *A. invadans* were maintained as positive and negative controls. Each treatment and control was performed in triplicate. 

### 2.5. Collection of Blood from the Fish and Separation of Serum

Five fish from each treatment were randomly selected for the collection of blood samples for biochemical analysis. Briefly, the fish were anesthetized with clove oil (@ 50 µL per liter water), and 2 mL hypodermal syringe and 24 gauge needles were used to collect blood (100–200 µL/ fish) by puncturing the caudal vein of the fish. The blood was kept in 1.5 mL Eppendorf tube without anticoagulant and stored in a freezer (4 °C) overnight. Afterward, the samples were centrifuged at 4000× *g* for 10 min at 4 °C and straw-colored serum was collected and stored in sterile Eppendorf tubes at −20 °C, until further used for analysis. All the procedures were carried out in sterilized conditions. 

### 2.6. Serum Biochemical Analysis

#### 2.6.1. Antioxidant Enzymes

The activity of antioxidant and metabolic enzymes was measured in serum following the standard techniques with slight modifications. Superoxide dismutase (SOD) activity was measured according to Misra and Fredovich (1972) in a medium containing sodium carbonate buffer (pH 10.2), EDTA, enzyme extract, and epinephrine. The change in absorbance was monitored at 480 nm using a microplate reader (BioTekEpoch^TM^2 Take-3 plate reader, California, USA). 

The method of Calibrone (1985) was used to determine catalase (CAT) activity. The drop in intensity at 240 nm was used to determine the breakdown of H_2_O_2_. The reaction solution comprised 50 mM phosphate buffer (pH 7.2) and 50 mM H_2_O_2_, and the efficiency of the reaction was determined at 240 nm using a microplate reader (BioTekEpoch^TM^2 Take-3 plate reader, USA) calibrated to 320 nm H_2_O_2_ an extinction coefficient of 40 M^−1^ cm^−1^. CAT units were defined as one unit of H_2_O_2_ decomposed per minute per milligram of protein.

#### 2.6.2. Serum Biochemical Indices and Immune Stress Responses

Serum protein was measured by using an automated biochemical analyzer (Auto Analyzer, Transasia-Erba EM–200, USA). The heat shock protein 70 (HSP70) and 90 (HSP70) activity in serum were quantified using standard protocols and reagents provided by the manufacturer. These assays were performed using an ELISA assay kit (EA0011FI and E0064FI, BT BioAssay, Shanghai, China), as per the manufacturer’s instruction. The final OD was taken at 450 nm using a microplate reader (BioTekEpoch^TM^2 Take-3 plate reader). The assay was performed in triplicate and is representative of two independent experiments. 

For the cortisol assay commercial kit provided by Bioassay technology laboratory (EA0004FI, BT BioAssay, Shanghai, China). Briefly, 20 μL of each of the cortisol standard solutions (0, 20, 50, 100, 200, 400 and 800 ng.ml^−1^) and fish serum samples were added in duplicate to the microplate. Similarly, the samples for the recovery and linearity test were dispensed in other wells. Standard solutions and fish serum samples for the recovery test were assayed in duplicate. Subsequently, 200 μL of enzyme conjugated to horseradish peroxidase was added into each well. Later, the wells were gently mixed for 10 min and incubated for 1 h at room temperature. The solution of each well was removed by washing the plate three times with 400 μL of PBS and shaking out the content onto absorbent paper to remove residual drops that could affect the accuracy and precision of the assay. Subsequently, 100 μL of TMB (tetramethyl benzidine) enzyme substrate was added to each well and incubated for 15 min at room temperature. The enzymatic reaction was visualized by the color change and was stopped by the addition of 100 μL of 0.5 M phosphoric acid (H_2_PO_3_). The intensity of color is inversely proportional to the concentration of cortisol in the samples. Absorbance was reading a spectrophotometer at 450 nm on a microtiter plate reader within 10 min after the addition of the stop solution. 

The total immunoglobulin M (IgM) in fish serum was measured using a commercial ELISA kit following the manufacturer’s instructions (E0025Fi, BT BioAssay, Shanghai, China). In brief, 50 μL of the standard sample was added on a standard well containing biotinylated antibody. Afterward, 40 μL of the sample, 10 μL anti-COR antibody, and 50 μL streptavidin-HRP were added into wells. The solution was thoroughly mixed and covered with plate sealer. The plate was incubated for 60 min at 37 °C. Later, the cover was removed and washed 5 times with wash buffer. During each washing, a minimum 0.35 mL of wash buffer was kept for 30 s to 1 min. An amount of 50 μL of substrate solution A and 50 μL of substrate solution B were added into wells. The plates were sealed and inoculated for 10 min at 37 °C in dark conditions. Afterward, 50 μL of stop solution was added in each well, and we observed the color change from blue to yellow. We measured the OD of each well immediately using a microplate reader at 450 nm within 10 min after the addition of the stop solution. 

### 2.7. Statistical Analysis

The data were expressed as mean ± SEM and arcsine transformed to satisfy normal distribution and homoscedasticity requirements as necessary. All other data were compared with one-way ANOVA followed by Bonferroni-type post hoc tests. Statistical analysis was performed using the IBM statistical software SPSS (version 24.0, IBM Corp., Armonk, NY, USA). The significance level was set at *p* values ≤ 0.05. 

## 3. Results

### 3.1. Eclipta alba Leaf Extract Protects H. fossilis against subsequent A. invadans Challenge

The fingerlings exposed to *Eclipta alba* methanolic leaf extract in the range of 5 to 100 ppm did not exhibit any significant difference in survival when compared with unexposed, control larvae (Figure 5A). In contrast, the 125 ppm extract induced significantly high toxicity, and 40 % mortality was observed in the treatment group. This result indicates that methanolic leaf extract appeared to be toxic in high concentrations under the present experimental conditions. In the next experiment, we examined the optimum dose of *A. invadans* required for the dose–response study. We found that *H. fossilis* fingerlings, when exposed to five infectious doses of *A. invadans* (0.5, 1, 2, 4, and 8 mL), showed a significant difference in the survival of fingerlings. The *A. invadans* infectious dose, 4 mL suspension of 100+ motile zoospores/litre of water, provoking significant mortality near 60% at 168 h, was considered an optimum dose and selected for the experimental challenge (Figure 5B). We next investigated whether E. alba methanolic leaf extract could confer protection to the host *H. fossilis* against *A. invadans*, which can cause epizootic ulcerative syndrome (EUS) in both freshwater and brackish water fish. We found that *H. fossilis* fingerlings that received leaf extract in the range of 5–100 ppm exhibited a significant increase in survival compared to positive control. However, the maximum survival (~2 folds) was observed at a concentration between 50–100 ppm (Figure 5C). Taken together, the results showed that the *E. alba* leaf methanolic extract plays an essential role in protecting the *H. fossilis* fingerlings against pathogenic *A. invadans*. 

### 3.2. Eclipta alba Leaf Extract Reduces Stress and Enhances Antioxidant Defense of H. fossilis Fingerlings

We next sought to investigate the mechanism of the action of *E. alba* leaf methanolic extract in inducing resistance against *A. invadans* challenge. The findings revealed that cortisol concentration was at a basal or resting level in the negative control group (NC) of fingerlings. Moreover, significant positive correlations between *A. invadans* Oomycetes infection and cortisol levels were observed in *H. fossilis*. The positive control group (PC) of fingerlings has a significantly higher levels of cortisol in the serum. Interestingly, methanolic leaf extract treatment significantly decreased the cortisol level and the lowest values were observed at a concentration between T4–T6 treatments (50–100 ppm) (Figure 6A). 

The activity of SOD and CAT was significantly elevated following *E. alba* leaf methanolic extract treatment at 168 h following the experimental challenge. The highest values were recorded in treatment T4–T6 groups supplemented with 50–100 ppm methanolic leaf extract (Figure 6B,C). Moreover, significantly lower values were reported in the PC group of fingerlings challenged with *A. invadans*, as compared with the NC group. These results indicate that reduced stress and enhanced antioxidant response generated by methanolic leaf extract appeared to be, at least in part, involved in the induction of protective immunity within the *H. fossilis* fingerlings. 

### 3.3. Supplementation of Eclipta alba Leaf Extract Enhances Specific and Non-Specific Immune Response of H. fossilis

Improved resistance due to plant-based compound mediation against microbial pathogens has been linked with enhanced immune response in fishes [20,22,23]. To gain insight into this, we used enzymatic assays to assess the role of non-specific (HSP70, HSP90) and specific (IgM) arms of immunity in protecting *H. fossilis* fingerlings against *A. invadans*. We found that *H. fossilis* fingerlings treated with *E. alba* leaf methanolic extract showed a significant increase in the HSP70 and HSP90 activity (Figure 7A,B). The maximum activity (~2 folds in both HSP70 and HSP70) was observed from the T5 group onwards (75 ppm extract concentration). Furthermore, our analysis revealed that methanolic leaf extract treatment to fingerlings showed a significant difference in the total protein level against *A. invadans* (Figure 7C). The protein concentration was significantly increased in the treated fingerlings, and the highest values were recorded in the T6 group supplemented with 100 ppm extract concentration. We also further examined the secreted immunoglobulins concentration, possibly induced by methanolic leaf extract by analyzing IgM activity in the serum of *H. fossilis* fingerlings. We found that treated fingerlings have significantly higher IgM activity in serum when compared with the negative and positive control fingerlings (Figure 7D). In parallel with non-specific immune parameters, higher IgM activity was observed from the T5 group onwards. Taken together, these results imply that *E. alba* leaf extract enhances both non-specific and specific immunity, which, in turn, leads to increased survival of *H. fossilis* fingerlings against Oomycetes *A. invadans* infection.

## 4. Discussion

The natural compounds/molecules confer protection and/or enhance immune reactivity against biotic and abiotic stressors in a manner conceptually equivalent to probiotics, vaccines, or immunostimulants [21,23,24,25,26]. These compounds have received great attention in recent years with regard to fish because of the additional biodegradable and biocompatible properties that work on the One Health approach, particularly relevant in food safety and combating antibiotic resistance [12,27,28]. The *Eclipta alba*, a medicinal herb belonging to Asteraceae, is widely available and distributed throughout India [29]. This plant extract has been reported to possess immunostimulatory properties and reported to enhance the phagocytic index, antibody titer, and WBC count in higher vertebrates. Interestingly, the *E. alba* extract has shown promising results in aquaculture animals. A study by Citarasu et al. [30] demonstrated that *E. alba* significantly enhances survival and reduced the viral load in shrimps. Furthermore, the extract supplementation has been reported to enhance both the humoral and cellular non-specific immune responses and provide disease resistance in *Oreochromis mossambicus* against *A. hydrophila* [31]. Given the potential of the plant-based compound *E. alba* as an anti-infective agent, understanding the underlying mechanism of action of this compound against fungal infection is considered vital.

The ability to stimulate the defense system of vertebrates is considered to be central to controlling the pathogenesis of many microbial pathogens [32,33]. In the case of aquatic animals that are constantly exposed to microbial pathogens, non-specific and specific immune factors might be a potential preventive modality for the control of biotic and abiotic stressors in the aquaculture production systems [34,35]. Although previous works have shown that natural/plant-based compounds increase the resistance of the host against bacterial infection by facilitating enhanced immune response [36,37], the role of natural/plant-based compounds against EUS-causing *A. invadans* Oomycetes pathogens and the molecular mechanism behind the protective effect in hosts remains to be established. In this study, using a *H. fossilis* and *A. invadans* host–pathogen model system, it was shown that *E. alba* leaf methanolic extract protects *H. fossilis* fingerlings against *A. invadans*.

The stress-mediated impairment of immune function has been widely described in cultured and wild fish and is associated with increased susceptibility to disease [38,39]. The stress response is initiated and controlled by the production of corticosteroids (mainly cortisol, a steroid hormone with many biological activities, including gluconeogenesis and immunosuppression). Cortisol, a principal corticosteroid released as part of the primary stress response, is normally associated with stressful and disease conditions, where the fish can perceive a threat. It plays a critical role in mediating adaptive metabolic, physiological, and behavioral adjustments. The level of cortisol in fish indicates the health status of animals, as extended increased cortisol level is often negatively associated with growth, development, disease resistance, immunity, and reproduction [34]. It is important to mention that oxidative stress and infectious disease are interrelated and the emergence of one phenomenon leads to the development of another and vice versa. For instance, the excessive production of reactive oxygen species (ROS) and reactive nitrogen species (RNS) by activated immune cells during microbial disease creates a highly cytotoxic environment that leads to an imbalanced immune response and direct damage to target organs [40]. In parallel, the study showed that *A. invadans* infection significantly induced stressful conditions in *H. fossilis* fingerlings, resulting in higher cortisol and lower antioxidative enzyme activity in the positive control group of fish. Interestingly, the observed cortisol concentrations are very low and indicate, at most, very mild stress. Similar observations were reported by Webster et al. (2020) in juvenile Atlantic salmon. Results showed that exposure to a mild confinement stressor for two weeks has a significant impact on the physiology of fish, marked by increased levels of cortisol [41]. In contrast, *E. alba* leaf methanolic extract supplementation exhibited a protective effect in *H. fossilis* fingerlings against *A. invadans* infection by a possible mechanism of anti-stress and antioxidant activity. In the methanolic extract treatment groups, reduced cortisol and enhanced antioxidant enzyme activity (SOD and CAT) were observed. The antioxidant enzymes, viz., superoxide dismutases (SOD) (detoxifies toxic superoxide anion radicals) and catalase (CAT) (primary antioxidant defense component), located within different cellular compartments (mitochondrial and cytosolic) in fish, are responsible for the generation of antioxidant defense response and imparting protection from oxidative stress by the detoxification of free radicals. Our results were consistent with the known beneficial roles of plant-based compounds to induce resistance in the host and prevent microbial infection functionally through anti-stress, antioxidant, and immunostimulatory properties [42,43].

There exists a correlation between the reduced stress and enhanced antioxidative response, with non-specific immune response (HSP70, HSP90) and specific immune response (IgM) of animals against microbial infection [20,21,42,43,44,45]. Interestingly, *E. alba* leaf methanolic extract significantly enhances HSP70, HSP90, and IgM levels in *H. fossilis* fingerlings against *A. invadans* infection. Our results indicate that increased survival in *H. fossilis* treated with leaf methanolic extract is mostly due to a significant improvement in the health status of the fish. This is due to the fact that apart from reducing cortisol levels and enhancing antioxidative response, the methanolic leaf extracts significantly induce both the non-specific and specific immunity of *H. fossilis* fingerlings.

## 5. Conclusions

The results presented here provide new insight into the immunomodulatory properties of *E. alba* methanolic leaf extract. Our study provides strong evidence that the initial generation of anti-stress, antioxidative response, and non-specific and specific immunity plays a key role in the protection of *H. fossilis* fingerlings against *A. invadans* infection. The ability of the *E. alba* leaf extract to boost immunity and induce resistance in *H. fossilis* makes it a potent biocontrol agent that may be valuable to avert *A. invadans* infection in other fish species. The results obtained add new information about the mode of action of *E. alba* methanolic leaf extract and advance our knowledge of this compound as a potential disease-mitigating agent.

## Figures and Tables

**Figure 1 jof-09-00142-f001:**
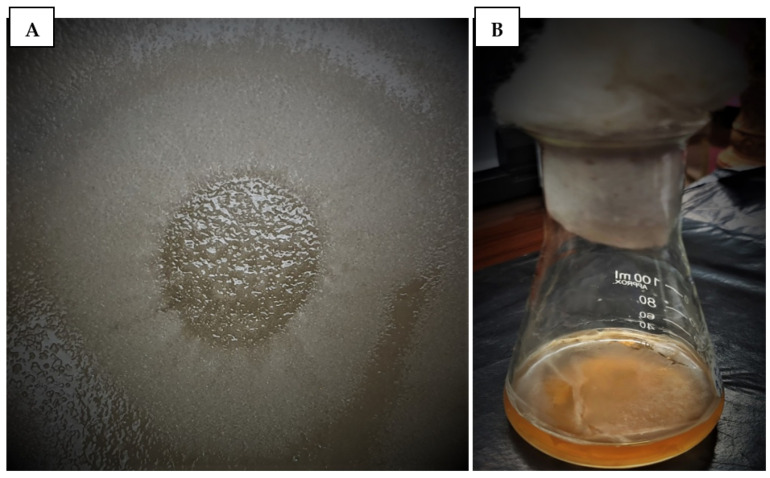
*Aphanomyces invadans* culture used in fish challenge experiments. (**A**) The fungal isolate was grown in potato dextrose agar (PDA) plates. (**B**) *A. invadans* cultured in potato dextrose broth (PDB) were used for different analyses.

**Figure 2 jof-09-00142-f002:**
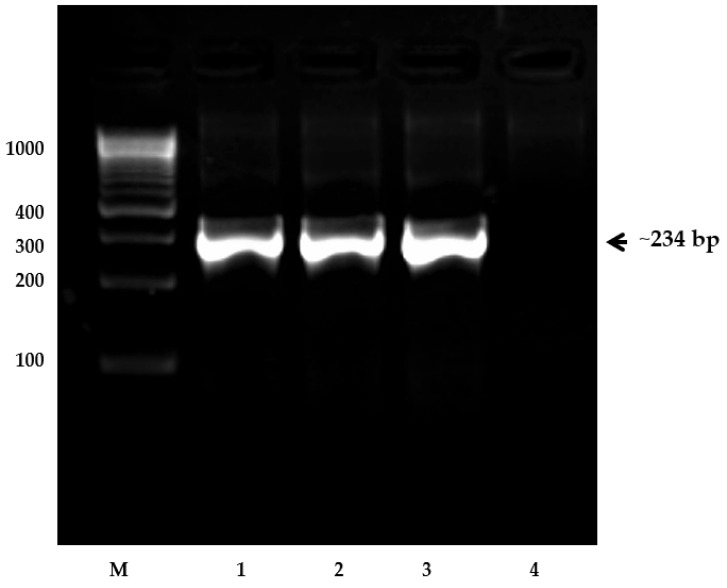
Agarose gel of PCR amplicon for identification of *Aphanomyces invadans* using ITS1 region primers. Lane 1–3: Positive amplicons of *A. invadans*; Lane 4: *Vibrio cholera* (EMM1) serotype (control). Molecular mass standards (M) 100 bp (DNA ladder) are shown on the left. Positive amplicon with arrow mark (~234 bp) is indication for pathogenic *A. invadans* identified from 1–3 isolate template DNA.

**Figure 3 jof-09-00142-f003:**
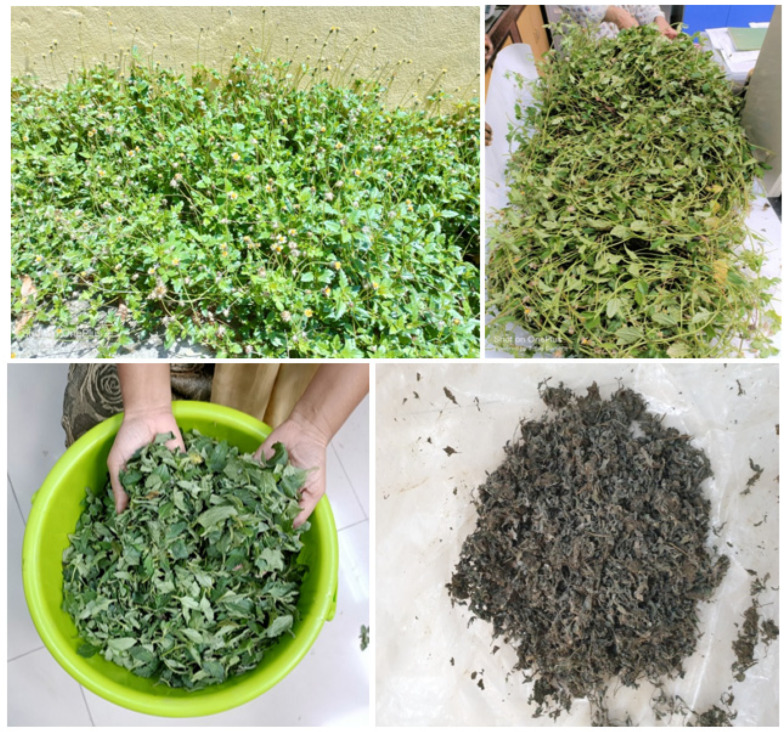
*Eclipta alba* (Bhringaraj) leaves used in the isolation of methanolic extract were collected from ICAR-CIFRI garden.

**Figure 4 jof-09-00142-f004:**
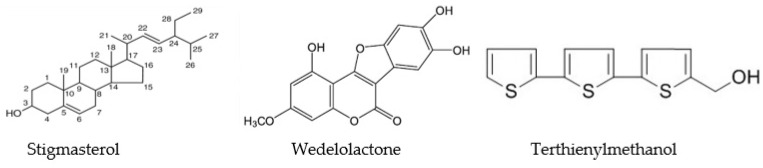
Chemical structure of *Eclipta alba* (Bhringaraj) extract bioactive compounds.

**Figure 5 jof-09-00142-f005:**
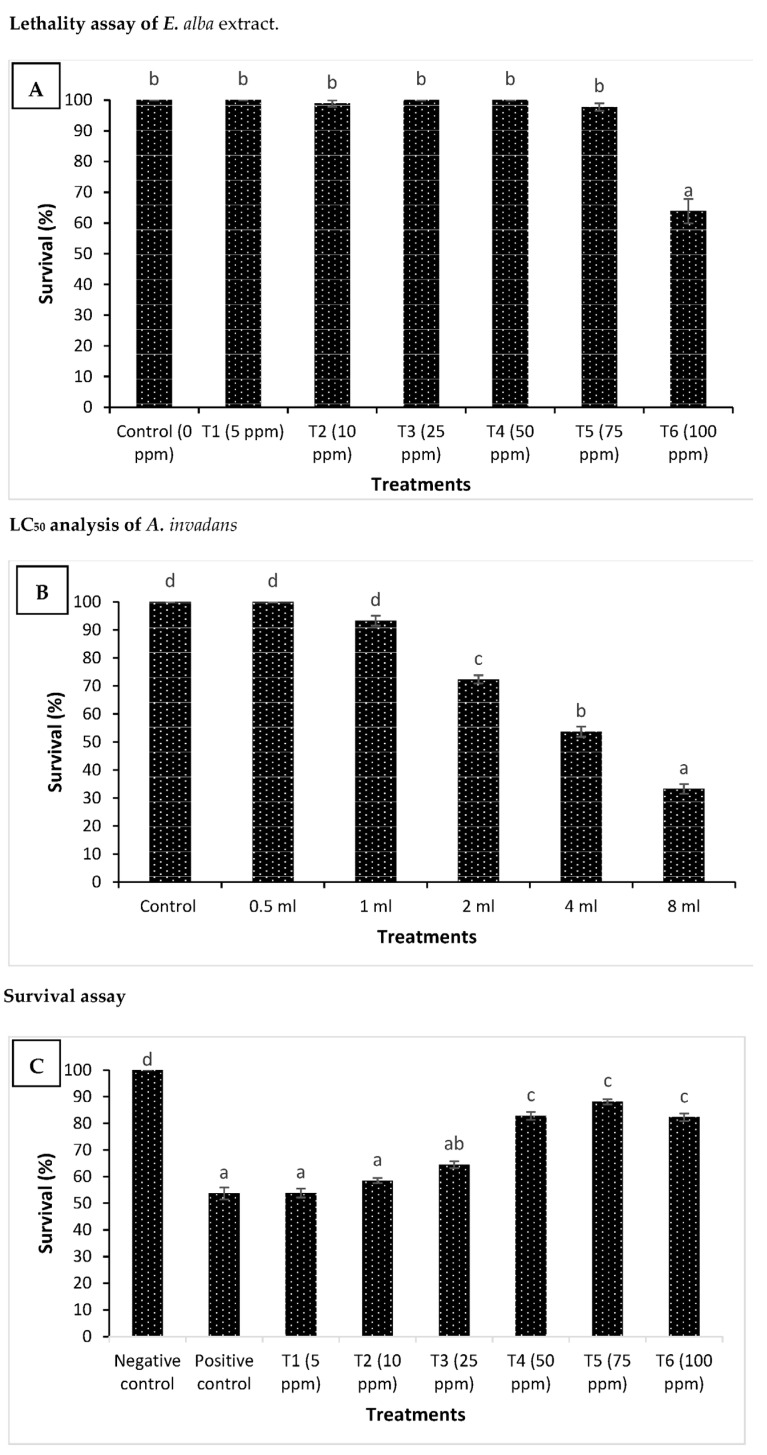
Eclipta alba (Bhringaraj) methanolic leaf extract treatment protects *Heteropneustes fossilis* fingerlings against *Aphanomyces invadans* challenge. (**A**) Toxicity of methanolic leaf extract to *H. fossilis* fingerlings. The fingerlings were treated with extract at the indicated doses for 168 h. The non-treated fingerlings served as control. Survival was recorded 168 h post-treatment. Error bars represent the standard error of five replicates; different letters indicate significant differences (*p* < 0.05). (**B**) The proportion of survived *H. fossilis* fingerlings after 168 h post-challenge with *A. invadans*. Five infectious doses (0.5, 1, 2, 4, and 8 mL suspension of 100+ motile zoospores/litre of water) were tested in the *H. fossilis* fingerlings. The *A. invadans* dose provoking significant mortality near to 60% at 168 h in *H. fossilis* fingerlings was considered an optimum dose and was selected for the experimental challenge. The unchallenged served as control. Values are presented as mean ± SE (*n* = 3). (**C**) Survival (%) of *H. fossilis* fingerlings treated with methanolic leaf extract after 168 h of challenge with *A. invadans*. The fingerlings were treated with leaf extract at the indicated doses and subsequently challenged with *A. invadans* at 4 mL suspension of 100+ motile zoospores/liters of water. Non-pretreated larvae that were either challenged with *A. invadans* (positive control) or unchallenged (negative control) served as controls. Error bars represent the standard error of five replicates; different letters indicate significant differences (*p* < 0.05).

**Figure 6 jof-09-00142-f006:**
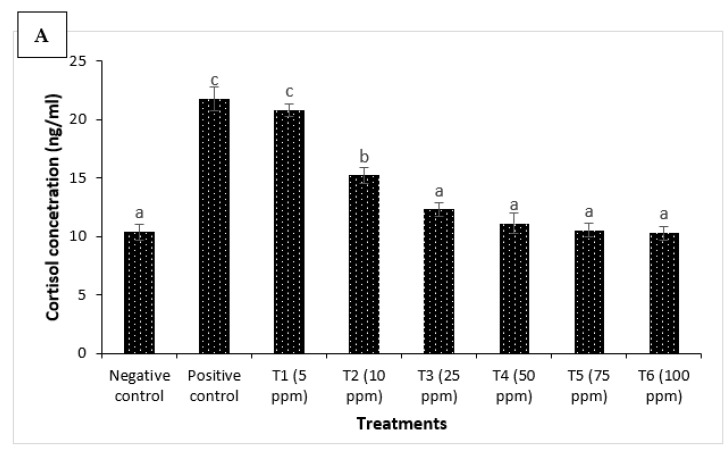
*Eclipta alba* leaf extract reduces stress and enhances antioxidant defense of *H. fossilis* fingerlings against *A. invadans*. The serum samples of *H. fossilis* fingerlings were collected from 8 treatment groups. The fingerlings in T1–T6 groups were treated with leaf extract at the indicated doses. Non-pretreated larvae that were either challenged with *A. invadans* (positive control) or unchallenged (negative control) served as controls. (**A**) cortisol concentration (ng/mL); (**B**) superoxide dismutase (SOD) activity (U/mg protein); (**C**) catalase (CAT) activity (U/mg protein). Error bars represent the standard error of five replicates; different letters indicate significant differences (*p* < 0.05).

**Figure 7 jof-09-00142-f007:**
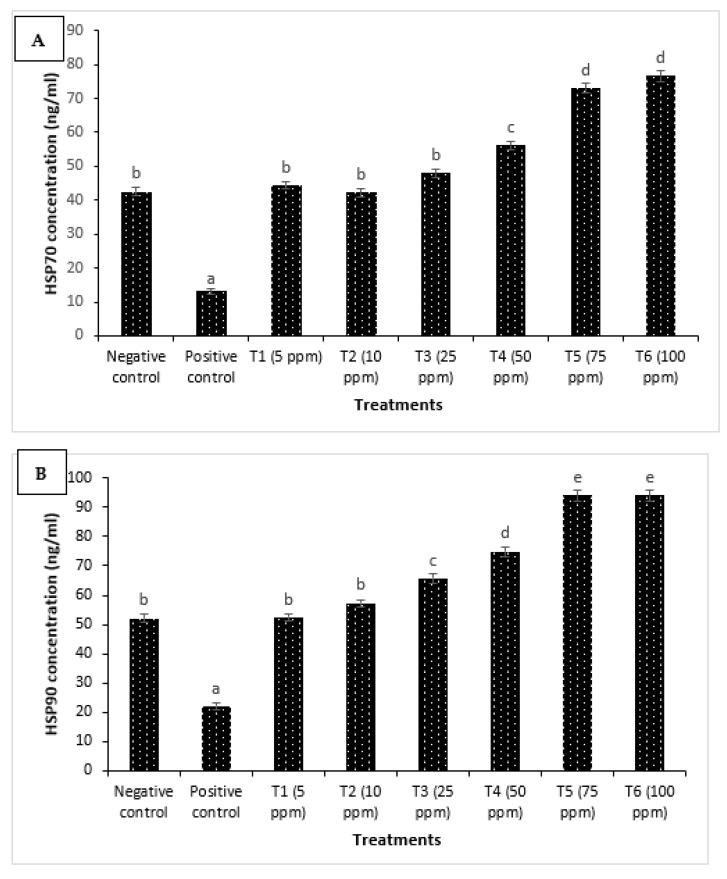
*Eclipta alba* leaf extract enhances both the specific and non-specific immune response of *H. fossilis* against *A. invadans*. The serum samples of *H. fossilis* fingerlings were collected from 8 treatment groups. The fingerlings in T1–T6 groups were treated with leaf extract at the indicated doses. Non-pretreated larvae that were either challenged with *A. invadans* (positive control) or unchallenged (negative control) served as controls. (**A**) HSP70 concentration (ng/mL); (**B**) HSP90 concentration (ng/mL); (**C**) protein level (mg/mL) and (**D**) IgM concentration (μg/mL). Error bars represent the standard error of five replicates; different letters indicate significant differences (*p* < 0.05).

**Table 1 jof-09-00142-t001:** Primers used in the confirmation of *A. invadens* isolate.

Target	Nucleotide Sequence		Fragment Size (bp)	Annealing Temperature (°C)
ITS1 region	Forward Reverse	5′-TCATTGTGAGTGAAACGGTG-3′ 5′-GGCTAAGGTTTCAGTATGTAG-3′	234	56

## Data Availability

The data presented in this study are available within the article.

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
