# Peer review of "Immunomodulatory Potency of Eclipta alba (Bhringaraj) Leaf Extract in Heteropneustes fossilis against Oomycete Pathogen, Aphanomyces invadans"

_jof, 2023, doi:10.3390/jof9020142_

Round 1

Reviewer 1 Report

The article title and content are completely consistent. The explanations and references in the text are consistent and sufficient. The methods used are explained in detail and clearly. Results; clearly express the effectiveness of the plant-based extract applied. Tables and sub-information/explanations of tables are understandable and well-written.

As a suggestion I can make to the authors, I think that it would be good to calculate the efficiency of the extraction method (i.e. the amount of plant product used and the amount of the final product), how long the obtained product can be used, and the potential economic benefit by making an approximate cost calculation.

Author Response

We sincerely thank the thorough review and helpful suggestions from the reviewer, which we believe have contributed to improving the manuscript. We have considered and tried to address all the comments and suggestions given by the reviewer.

Furthermore, we have worked on improving the manuscript abstract, methods section, figures and tables. We sincerely hope that the revised version of the manuscript will meet the scientific rigor, and journal standards and will be considered by the Journal of Fungi.  

We have attempted to answer the comments systematically. In the following paragraphs, you will find our response to the comments.

  1. As a suggestion I can make to the authors, I think that it would be good to calculate the efficiency of the extraction method (i.e. the amount of plant product used and the amount of the final product), how long the obtained product can be used, and the potential economic benefit by making an approximate cost calculation. 
  • We included the extraction method in the revised manuscript.
  • Although we don’t have data on the shelf life of the product, we could see the immunomodulatory properties in 1 year old extract stored at 4 0

Reviewer 2 Report

This work is important by providing evidence that an oomycete-caused disease problems could be lessened with plant-based chemicals. The manuscript is generally rather well written, but some parts require important clarifications and revisions before the work can be published. Most importantly, the authors do not actually conclusively demonstrate what they claim: "We further demonstrated that the A. invadans-protective effect of methanolic leaf extract was caused by its immunomodulatory effect and is linked to the enhanced survival of fingerlings." as they demonstrate a correlation, not causality. As far as I understand the methods, the authors simultaneously exposed the fish to the oomycete and the leaf extact. In order to demonstrate causality, the authors should first expose the fish to leaf extract, then demonstrate an immunological change that improves survival without any leaf extract. If the experiment was done like this, it is not clear from what is written. This must be clear before this manuscript can be accepted. If the exposure was simultaneous to both the oomycete and the leaf extract, all responses can be caused by either or both of the stressors.

Second, the authors claim they studied innate and specific immunity. Heat shock proteins reflect generic stress responses, not innate immunity. Actually the authors do not have any clear measures of innate immunity. IgM levels reflect generic levels of antibodies, not specific response to A. invadans, at least from what I understand from the methods. If the authors first challenged the fish with A. invadans and measured the specific response to A. invadans, the use of specific immunity term is correct (but not clear from the methods). Simple measurement of IgM levels measures general level of antibodies, and the authors should make clear they did not measure specific immune response but just general level of IgM type antibodies which can reflect both past infection history and capacity to form antibodies.

The description of the use of ELISA to measure cortisol levels is not clear. It is not said what kind of cortisol antibody was used.

The description of the statistical analyses (2.7.) is not clear. It is unclear how arcsin transformation can be used and how it improves the distributions. Generally, arcsine squareroot transformation is applied for proportions (like survival per tank). The statistical unit is unclear and the individual tests are not detailed. It is unclear why authors chose alpha level 0.001 instead of conventional 0.05. There is no mentioning how Bonferroni issue was handled. The statistics must be completely revised.

Importantly, the authors not provide any ethical statement while they conducted ethically problematic infections that killed some of the fish. The authors must explain why they could not follow the development of infection without letting the disease to kill the fish. Word fingerling is not scientific, please specify the age of the fish used.

Minor / detailed comments:

Abstract:

"Taken together, our study provides substantial evidence that 29 the generation of anti-stress, antioxidative response and non-specific and specific immunity plays 30 important role in protecting H. fossilis fingerlings against A. invadans infection." Correct: "Taken together, our study provides evidence that the generation of anti-stress and antioxidative responses, and humoral immunity may play roles in protecting H. fossilis fingerlings against A. invadans infection."

Oomycete is not a fungus. Revise the wording throughout the manuscript.

Line 50: Instead of the history, please introduce the measures you use for stress reaction and humoral immunity.

Line 82: Make a clear aims and questions paragraph and list here all the response variables you measured.

Line 172: Fishes = several fish species. Multiple fish individuals = many fish

Line 179: How long was the exposure?

Line 198: How long was the exposure?

Line 208: Not clear if the tanks had oomycete spores and leaf extract at the same time. This is crucial for the interpretation of the results.

Line 221: How much blood was collected?

Line 222: Why? Why you used serum and not plasma?

Line 250: So you used self-made ELISA for cortisol measures? What was the antibody and how it was produced?

Line 265: Make clear if you analysed specific IgM to the oomyce or just all IgM.

Line 285: Explain the methods in Methods and present only results in Results

Line 306 / figure: Survival cannot exceed 100%

Figure 5 B: Give zoospore densities, not ml:s.

Lines 330-356: This information should go into Introduction and Methods

Line 368: HSPs are not immunological proteins but general antistress proteins

Line 416: Repeat, delete. All information of this sort should go to Introduction.

Line 460: Is most likely. Please do not assume, explain how you can justify this conclusion. I think your positive and negative controls can help in the interpretation but you need to better introduce the design and help the reader to interpret the results based on these. Currently, the discussion is too general and light compared to the results.

Line 481: Why is this not applicable? Is there something that prevents you from publishing your data?

Author Response

We sincerely thank the thorough review and helpful suggestions from the reviewer, which we believe have contributed to improving the manuscript. We have considered and tried to address all the comments and suggestions given by you and the reviewer.

Furthermore, we have worked on improving the manuscript abstract, methods section, figures and tables. We sincerely hope that the revised version of the manuscript will meet the scientific rigor, and journal standards and will be considered by the Journal of Fungi.  

We have attempted to answer the comments systematically. In the following paragraphs, you will find our response to the comments.

  1. In order to demonstrate causality, the authors should first expose the fish to leaf extract, then demonstrate an immunological change that improves survival without any leaf extract. If the experiment was done like this, it is not clear from what is written. This must be clear before this manuscript can be accepted. If the exposure was simultaneous to both the oomycete and the leaf extract, all responses can be caused by either or both of the stressors.
  • We completely agreed with the reviewer comments that simultaneous exposure of both the oomycete and the leaf extract might have an effect of compound and oomycete interaction. However, to minimize this we kept a positive control (only oomycete) in each assay.

  1. The authors claim they studied innate and specific immunity. Heat shock proteins reflect generic stress responses, not innate immunity. Actually the authors do not have any clear measures of innate immunity. IgM levels reflect generic levels of antibodies, not specific response to A. invadans, at least from what I understand from the methods. If the authors first challenged the fish with A. invadans and measured the specific response to A. invadans, the use of specific immunity term is correct (but not clear from the methods). Simple measurement of IgM levels measures general level of antibodies, and the authors should make clear they did not measure specific immune response but just general level of IgM type antibodies which can reflect both past infection history and capacity to form antibodies.
  • We completely agree with the reviewer comments. However, in several papers, Hsps are considered important molecules and have a significant role in immunity and disease resistance (Srivastava 2002 Nature Reviews Immunology; Pockley and Henderson 2017 Philosophical Transactions of the Royal Society B; Kumar et al., 2018 Fronters in immunology; Krüger et al., 2019 Journal of Applied Physiology). Hence, in this study we studied the role of methanolic in Heteropneustes fossilis Hsps activity against Aphanomyces invadans.
  • We have analyzed total IgM in fish serum and not specific IgM to the oomycete, accordingly text has been modified

  1. The description of the statistical analyses (2.7.) is not clear. It is unclear how arcsin transformation can be used and how it improves the distributions. Generally, arcsine squareroot transformation is applied for proportions (like survival per tank). The statistical unit is unclear and the individual tests are not detailed. It is unclear why authors chose alpha level 0.001 instead of conventional 0.05. There is no mentioning how Bonferroni issue was handled. The statistics must be completely revised.
  • We appreciate and agree with the comment that we couldn’t able to justify clearly explaining the statistical analysis. Accordingly, the statistical analysis is modified in the revised manuscript.

  1. Importantly, the authors not provide any ethical statement while they conducted ethically problematic infections that killed some of the fish. The authors must explain why they could not follow the development of infection without letting the disease to kill the fish. Word fingerling is not scientific, please specify the age of the fish used.
  • As suggested, the ethical statement is included in the revised manuscript.

Minor comments

Abstract:

"Taken together, our study provides substantial evidence that the generation of anti-stress, antioxidative response and non-specific and specific immunity plays important role in protecting H. fossilis fingerlings against A. invadans infection."

  • As suggested, the text has been modified. "Taken together, our study provides evidence that the generation of anti-stress and antioxidative responses, and humoral immunity may play roles in protecting H. fossilis fingerlings against A. invadans infection."

Oomycete is not a fungus. Revise the wording throughout the manuscript.

  • The text has been revised accordingly.

Line 50: Instead of the history, please introduce the measures you use for stress reaction and humoral immunity.

  • As suggested by the reviewer, the text has been modified.

Line 82: Make a clear aims and questions paragraph and list here all the response variables you measured.

  • As per the suggestion, the paragraph has been modified.

Line 172: Fishes = several fish species. Multiple fish individuals = many fish

  • As suggested, the text has been modified.

Line 179: How long was the exposure?

  • As suggested, the text is included.

“Line 198: How long was the exposure?

  • As per the suggestions, the sentences have been modified.

Line 208: Not clear if the tanks had oomycete spores and leaf extract at the same time. This is crucial for the interpretation of the results.

  • The oomycete spores and leaf extract were added simultaneously, accordingly text has been modified.

Line 221: How much blood was collected?

  • As per the suggestions, the details have been included.

Line 222: Why you used serum and not plasma?

  • In general, serum samples are most widely used in research. Additionally, most standards for the assay are based on serum but not plasma. Hence, we took serum samples for analysis.

Line 250: So you used self-made ELISA for cortisol measures? What was the antibody and how it was produced?

  • As suggested by the reviewer, details have been included in the text.

Line 265: Make clear if you analysed specific IgM to the oomycete or just all IgM.

  • We have analyzed total IgM in fish serum and not specific IgM to the oomycete, accordingly text has been modified.

Line 285: Explain the methods in Methods and present only results in Results

  • As suggested by the reviewer, the text has been modified.

Line 306 / figure: Survival cannot exceed 100%

  • As per the suggestions, the figures have been modified

Figure 5 B: Give zoospore densities, not ml:s.

  • As per the suggestion, the figure legend has been modified.

Lines 330-356: This information should go into Introduction and Methods

  • As suggested, the text has been moved from results to the discussion.

Line 368: HSPs are not immunological proteins but general antistress proteins

  • We completely agree with the reviewer comments. However, in several papers, Hsps are considered important molecules and have a significant role in immunity and disease resistance (Srivastava 2002 Nature Reviews Immunology; Pockley and Henderson 2017 Philosophical Transactions of the Royal Society B; Kumar et al., 2018 Fronters in immunology; Krüger et al., 2019 Journal of Applied Physiology). Hence, in this study we studied the role of methanolic in Heteropneustes fossilis Hsps activity against Aphanomyces invadans.

Line 416: Repeat, delete. All information of this sort should go to Introduction.

  • As suggested by the reviewer, the text has been modified.

Line 460: Is most likely. Please do not assume, explain how you can justify this conclusion. I think your positive and negative controls can help in the interpretation but you need to better introduce the design and help the reader to interpret the results based on these. Currently, the discussion is too general and light compared to the results.

  • We agreed with the reviewer comments. We have tried to improve the discussion to make it more interesting and informative.

Line 481: Why is this not applicable? Is there something that prevents you from publishing your data?

  • As suggested by the reviewer, we have rephrased the text.

Round 2

Reviewer 2 Report

The authors have improved some parts of the manuscript, but some concerns remain. The new text additions must be proof-read for correct English. For the replicability of the work it would be essential that the authors give detailed information for the commerical kits they have used. None of the BT Lab cortisol kits matches with the information given by the authors: https://www.bt-laboratory.com/index.php/Shop/Index/productList.html?kw=cortisol and no information is given for the manufacturer of the IgM kits. Generally, the observed cortisol concentrations are very low and indicate at maximum very mild stress. This could be discussed better under the fish stress literature.

Still, no information is given why P < 0.001 is used as alpha level instead of standard 0.05 and the description of statistical methods is very general. No F-test results are provided in the manuscript despite the authors claim they used ANOVA. I suggest adding the F-stats and using standard alpha 0.05 but Bonferroni-type post hoc tests.

Now, it is said that the data are provided within the article but this is not true. No raw data are shown and cannot be shown directly in the article. Because the figures look nearly too good to be true, it would add credibility for the work, if the authors published their raw data.

Finally, the first author has cited several his own works but not all shrimp works are relevant for this article in finfish. This thus appears as unethical way to increase own citation metrics, and self-citations should be reduced despite this research group does pioneering work in this field.

Author Response

We sincerely thank you for your response and comments. We have considered and tried to address all the comments and suggestions given by you and the reviewer.

Furthermore, we have worked on improving the manuscript abstract, methods section, figures and tables. We sincerely hope that the revised version of the manuscript will meet the scientific rigor, and journal standard and will be considered by Journal of Fungi.  

We have attempted to answer the comments systematically. In the following paragraphs, you will find our response to the comments.

  1. For the replicability of the work it would be essential that the authors give detailed information for the commerical kits they have used. None of the BT Lab cortisol kits matches with the information given by the authors: https://www.bt-laboratory.com/index.php/Shop/Index/productList.html?kw=cortisol and no information is given for the manufacturer of the IgM kits. Generally, the observed cortisol concentrations are very low and indicate at maximum very mild stress. This could be discussed better under the fish stress literature.
  • As suggested by the reviewer, we have included all the commercial kit details in the revised manuscript.

  1. Still, no information is given why P < 0.001 is used as alpha level instead of standard 0.05 and the description of statistical methods is very general. No F-test results are provided in the manuscript despite the authors claim they used ANOVA. I suggest adding the F-stats and using standard alpha 0.05 but Bonferroni-type post hoc tests.
  •  As suggested, we have done all the statistical analysis with Bonferroni-type post hoc tests using standard alpha 0.05. Accordingly, graphs have been modified in the revised manuscript.

  1. Now, it is said that the data are provided within the article but this is not true. No raw data are shown and cannot be shown directly in the article. Because the figures look nearly too good to be true, it would add credibility for the work, if the authors published their raw data.
  •  As suggested by the reviewer, we have included the data set as supplementary file.

  1. Finally, the first author has cited several his own works but not all shrimp works are relevant for this article in finfish. This thus appears as unethical way to increase own citation metrics, and self-citations should be reduced despite this research group does pioneering work in this field.
  •  As suggested, we have removed the non-relevant and own-cited papers from the revised manuscript